# A SELF-EXPLANATORY METHOD FOR THE BLACK BOX PROBLEM ON DISCRIMINATION PART OF CNN

## ABSTRACT

Recently, for finding inherent causality implied in CNN, the black box problem of its discrimination part, which is composed of all fully connected layers of the CNN, has been studied by different scientific communities. Many methods were proposed, which can extract various interpretable models from the optimal discrimination part based on inputs and outputs of the part for finding the inherent causality implied in the part. However, the inherent causality cannot readily be found. We think that the problem could be solved by shrinking an interpretable distance which can evaluate the degree for the discrimination part to be easily explained by an interpretable model. This paper proposes a lightweight interpretable model, Deep Cognitive Learning Model(DCLM). And then, a game method between the DCLM and the discrimination part is implemented for shrinking the interpretation distance. Finally, the proposed self-explanatory method was evaluated by some contrastive experiments with certain baseline methods on some standard image processing benchmarks. These experiments indicate that the proposed method can effectively find the inherent causality implied in the discrimination part of the CNN without largely reducing its generalization performance. Moreover, the generalization performance of the DCLM also can be improved.

## 1 INTRODUCTION

Convolution neural network(CNN) has surpassed human abilities in some specific tasks such as computer game and computer vision etc. However, they are considered difficult to understand and explain(Brandon, 2017), which leads to many problems in aspects of privacy leaking, reliability and robustness. Explanation technology is of immense help for companies to create safer, more trustable products, and to better manage any possible liability of them (Riccardo et al., 2018). Recently, for finding inherent causality implied in the CNN, the unexplainable problem of CNN, especially concerning the discrimination part which is composed of the fully connected layers of the CNN, has been studied by different scientific communities. Many methods were proposed, which can extract various interpretable models from the optimal discrimination part based on inputs and outputs of the part for expressing the inherent causality implied in the part. However, because of data bias and noisy data in the training data set, the inherent causality cannot readily be found because the part is difficult to be approximated by any interpretable model. We think that the problem could be solved by the following procedure. Firstly, a lightweight interpretable model is designed which can be easily understood by human. And then, the model is initiatively extracted from the discrimination part by solving a Maximum Satisfiability(MAX-SAT) problem based on the activated states of the neurons in the first layer and the output layer of the part. An new distance is proposed which can evaluate the degree to which the discrimination part is easily explained, namely as interpretability performance or interpretable distance. For shrinking the interpretable distance, a game process between the interpretable model and the discrimination part is implemented. Finally, the optimal interpretable model can be obtained, which can express inherent causality implied in the discrimination part. Moreover, based on the procedure, it is also possible to monitor the evolution of the inherent causality implied in the part in the game process.

Main contributions of this paper can be summarized as follows:

- An interpretable model, Deep Cognitive Learning Model(DCLM), is proposed to express the inherent causality implied in the discrimination part, and a greedy method is given

for initiatively extracting the DCLM from the discrimination part by solving its Maximum Satisfiability(MAX-SAT) Problem.

- A new game method is proposed to improve the interpretability performance of the discrimination part without largely reducing its generalization performance by iteratively shrinking the interpretable distance between DCLM and the discrimination part.

- A new distance is proposed to evaluate the degree to which the discrimination part is easily explained, namely as interpretability performance or interpretable distance.

## 2 RELATED WORK

There are usually two types of methods for the unexplainable problem of the discrimination part, such as post-hoc method and ante-hoc method (Holzinger et al., 2019). However, because ante-hoc method is a transparent modeling method(Arrietaa et al., 2020), it can not obtain an explanation about the discrimination part. So, the post-hoc method will be reviewed.

Early post-hoc method can obtain global explanations for a neural network by extracting an interpretable model. Some references(Craven & Shavlik, 1999; Krishnan et al., 1999; Boz, 2002; Johansson & Niklasson, 2009) proposed a few methods that can find a decision tree for explaining a neural network by maximizing the gain ratio and an estimation of the current model fidelity. Other references (Craven & Shavlik, 1994; Johansson & Niklasson, 2003; Augasta & Kathirvalavakumar, 2012; Sebastian et al., 2015; Zilke et al., 2016) proposed rule extraction methods for searching optimal interpretable rules from a neural network.

Recently, some feature relevance methods have become progressively more popular. Montavon et al.(Montavon et al., 2017) proposed a decomposition method from a network classification decision into contributions of its input elements based on deep Taylor decomposition. Shrikumar et al.(Shrikumar et al., 2016) proposed DeepLIFT which can compute importance scores in a multilayer neural network by explaining the difference of the output from some reference output in terms of differences of the inputs from their reference inputs.

Some other works make complex black box model simpler. Che et al.(Che et al., 2017) proposed a simple distillation method called Interpretable Mimic Learning for extracting an interpretable simple model by gradient boosting trees. Thiagarajan et al.(Thiagarajan et al., 2016) build a Treeview representation of the complex model by hierarchical partitioning of the feature space. In addition, some references (Hinton et al., 2015; Bucila et al., 2006; Frosst & Hinton, 2017; Traore et al., 2019) proposed the distillation method of knowledge from an ensemble of models into a single model. Wu et al.(M. Wu, 2018) proposed a tree regularization method via knowledge distillation to represent the output feature space of a RNN based on a Multilayered perception. However, these methods can only solve the unexplainable problem of trained neural network or trained deep neural networks with explicit input characteristics. Wan et al.(Wan et al., 2020) constructed a decision tree using the last fully connection layer of the discrimination part of a CNN based on a prior structure.

In the paper, our goal is to find the inherent causality implied in the discrimination part of CNN, which is composed of all fully connected layers of the CNN without hurting its generalization performance by initiatively extracting its logic relationships with no prior structure and finally obtain its explanation by these logic relationships.

## 3 DEEP COGNITIVE LEARNING MODEL

For expressing the causal relationship between these neurons in the discrimination part, a new interpretable model is designed in the section. As we all known, a CNN includes a feature extractor and a discrimination part. The feature extractor composes of some convolution layers and some pooling layers. The outputs from the feature extractor are the inputs of the discrimination part of the CNN, namely feature maps, $\tau_1, \tau_2, ..., \tau_k$ where $k$ is the number of feature maps. All these feature maps form a feature set $\Gamma$.

We suppose that the discrimination part should better be explained by the logic relationships of the activated states of the neurons in its first layer and its output layer. This is because the relationships

are the inherent property of the part. In order to express the relationships, a deep cognitive learning model (DCLM)is proposed, shown in Fig.1(b).

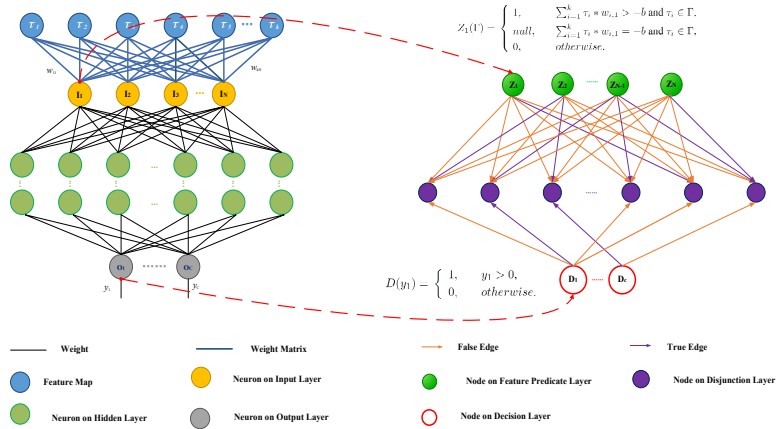

(a)Discrimination Part of CNN          (b)Deep Cognitive Learning Model(DCLM)

Figure 1: Discrimination part and DCLM of a CNN(A predication $Z_i$ indicates an activated state of a neuron $I_i$ of the discrimination part of the CNN and a predication $D_i$ indicates an activated state of a neuron $O_i$)

The DCLM consists of three layers:feature predicate layer, disjunction layer, and decision layer. The top layer is feature predicate layer which consists of many nodes. Every node has a predicate $Z_j(\Gamma)$ that expresses a positive action or negative action of features which the $j$th neuron in the first fully connected layer of the discrimination part captures. The predicate $Z_j(\Gamma)$ is defined as follows:

$$Z_1(\Gamma) = \begin{cases} 1, & \sum_{i=1}^{k} \tau_i * \boldsymbol{w}_{i,1} > -b \text{ and } \tau_i \in \Gamma, & (1) \\ null, & \sum_{i=1}^{k} \tau_i * \boldsymbol{w}_{i,1} = -b \text{ and } \tau_i \in \Gamma, & (1') \\ 0, & otherwise. & (1'') \end{cases}$$

where $j \in 1, 2, ..., N$, $N$ is the number of the input neurons of the first fully connection layer of the discrimination part of the CNN. $\boldsymbol{w}_{i,j}$ is a weight vector between the $i$th feature map and the $j$th neuron, $b_j$ is the bias of the $j$th neuron, and "$*$" is a convolution operation. "1" and "0" denote a positively activated state and a negatively activated state of the neuron respectively. "null" denotes an inactivated state.

The bottom layer is a decision layer which includes all nodes used for decision. Every node has a predicate which expresses an activated state of an output neuron of the discrimination part. It is defined as follows:

$$D(y_1) = \begin{cases} 1, & y_1 > 0, & (2) \\ 0, & otherwise. & (2') \end{cases}$$

where $i \in 1, 2, ..., C$,$C$ is number of the output neurons of the CNN, $y_i$ is the output value of the $i$th output neuron of the discrimination part. All nodes on the feature predicate layer and every node on the decision layer are connected to one or more nodes on the middle layer, namely as disjunction layer, with true or false edges. Every node represents a disjunction relation, which is expressed by a disjunctive normal form. It is worth mentioning: if a node is connected to a node on the disjunction layer by a false edge, its predicate follows after a non-operator in the disjunctive normal form.

The potential function of a disjunctive normal form can be obtained by using the Lukasiewicz method(Giles, 1975).

$$\phi_c(y_i) = \min(1, T(\Gamma, y_i)) \qquad (3)$$

where $T(\Gamma, y_i) = \sum_{j=1}^{N} \{a_j[1 - Z_j(\Gamma)] + (1 - a_j)Z_j(\Gamma)\} + (a_N + 1)D(y_i)$ and $N$ is the number of the nodes on the feature predicate layer. If $a_j = 1$, there is a false edge. Otherwise, there is a true edge.

The conditional probability distribution that a ground DCLM including all disjunctive normal forms is true is

$$p(\mathbf{y}, \Gamma) = \frac{1}{\Xi} \exp\left(\frac{\sum_{i=1}^{G} \lambda_i \phi_{ci}(y_i)}{\sum_{i=1}^{G} \lambda_i}\right) \tag{4}$$

where $G$ is the number of all ground formulas, $\Xi = \sum_{\Gamma \in \mathbb{F}} \exp\left(\frac{\sum_{i=1}^{G} \lambda_i \phi_{ci}(y_i)}{\sum_{i=1}^{G} \lambda_i}\right)$ is a partition function, $\mathbf{y} = (y_1, y_2, ..., y_G)$, $y_i$ is an output value of the CNN and $\lambda_i$ is a weighted value of the $i$th ground formula.

By maximizing its likelihood function, the optimal $a_i$ and $\lambda_i$ in the DCLM can be obtained.

$$C(\Gamma) = arg \max_{a_i, \lambda_i}[\log p(\mathbf{y}, \Gamma)] = arg \max_{a_i, \lambda_i}\left(\frac{\sum_i \lambda_i \phi_{ci}(y_i)}{\sum_i \lambda_i} - \log(\Xi)\right) \tag{5}$$

For extracting a optimal DCLM, a Maximum A Posterior(MAP) algorithm on the Maximum Satisfiability Problem (MAX-SAT) was designed. Using the disjunction normal form with the greatest weighted value in the optimal DCLM, a prediction of an input image can be obtained.

## 4 EVALUATION OF INTERPRETABILITY PERFORMANCE

We consider that if the discrimination part of a CNN has a similar shape of function curve with its optimal interpretable model, the former can be easily explained by the latter. Therefore, the interpretable performance of the discrimination part can be measured by the shape similarity between it and its optimal interpretable model. We posit that given the same input data set, the similarity may be measured by variance of differences between outputs of the both models. It can be named interpretation distance. It is easily proved that the smaller the interpretation distance is, the more similar their shapes are, and the better the interpretability performance of the discrimination part would be.

**Definition 1** If $X$ is a compact metric space and $\nu$ is a Borel measure in $X$, such as Lebesgue measure or marginal measures, in $\mathcal{L}_\nu^2(X)$, a square integrable function space on $X$, the interpretation distance, $\phi_d(P^*, f)$, between a discrimination part $f(x)$ and its optimal DCLM $P^*(x)$ is

$$\phi_d(P^*, f) = \int_Z (f(x) - P^*(x) - \mu^{P^*}(f))^2 d\nu \tag{6}$$

where

$$\mu^{P^*}(f) = \int_Z (f(x) - P^*(x)) d\nu \tag{7}$$

## 5 GAME BETWEEN A DCLM AND THE DISCRIMINATION PART OF A CNN

As discussed above,when the shapes of the discrimination part of a CNN and its optimal interpretable model are enough similar, the discrimination part has well interpretability performance. However, its generalization performance will tend to decrease. This is mainly attributed to the fact that because of data bias and noisy data in training data set, the sufficient and necessary condition for the consistent convergence of the two performances, $\phi_d(P^*, f^*) = 0(f^*$ is the optimal prediction model), is difficult to be guaranteed. Therefore, for the tradeoff problem, in training process, extracting an DCLM from the discrimination part and then iteratively reducing the interpretation distance between the two models may be a feasible solution. A detailed discussion about the problem can be found in the appendix.

To avoid reducing the generalization performance, the maximum probability $p(\mathbf{w} \mid X, \mathbf{y}_t)$ should be guaranteed, where $X$ is a training sample, $\mathbf{w}$ is the parameter set of the CNN and $\mathbf{y}_t$ is the target vector of $X$.

$$p(\mathbf{w} \mid X, \mathbf{y}_t) = \frac{p(\mathbf{w} \mid X)p(\mathbf{y}_t \mid \mathbf{w}, X)}{p(\mathbf{y}_t \mid X)} \propto p(\mathbf{w})p(\mathbf{y}_t \mid w, X) \tag{8}$$

---

**Algorithm 1** Game between DCLM and the discrimination part of a CNN(Its time complexity is $O(N + M)$ where $O(N)$ is a time complexity of training CNN,$O(M)$ is a time complexity of construction of Logic Net.)

---

**Input:** $Data\_set$
**Output:** $DCLM$
Repeat
$CNN = CNN\_Train(Data\_set, Adam, loss = CrossEntropy)$
**for** data,label **in** Data set **do**
   $Feature\_map = CNN\_feature_extractor(CNN).$
   $disjunctive\_normal\_form = Disjunction(Feature\_map, Fnn, W_{i,j}, Rule1 = Eq.1, Rule2 = Eq.2)$
   $UpData\_DCLM(disjunctive\_normal\_form, Updata = Eq.5)$
**end for**
**for** $i = 1$ **to** $n$ **do**
   **for** data,label **in** Data set **do**
      $Feature\_map = CNN\_feature\_extractor(CNN)$
      $ym = DCLM(Feature\_map)$
      $CNN\_DCLM(data, ym, loss = Eq.14)$
   **end for**
**end for**
Until the interpretation distance and accuracy converge

---

where $p(\mathbf{y}_t \mid \mathbf{w}, X) = \int p(\mathbf{y}_t \mid f, \mathbf{w}, X) \int p(f \mid \mathbf{y}_{dclm}, \mathbf{w}, X) p(\mathbf{y}_{dclm} \mid \mathbf{w}, X) d\mathbf{y}_{dclm} df$ and $\mathbf{y}_{dclm}$ is a prediction of DCLM.

When the DCLM is known, $\mathbf{y}_{dclm}^*$ is its optimal prediction and $p(\mathbf{y}_{dclm}^* \mid \mathbf{w}, X) = 1$. Then

$$\int p(f \mid \mathbf{y}_{dclm}, \mathbf{w}, X) p(\mathbf{y}_{dclm} \mid w, X) d\mathbf{y}_{dclm} = p(f \mid \mathbf{y}_{dclm}^*, \mathbf{w}, X) \tag{9}$$

Similarly, known the input $X$ and $\mathbf{w}$, $f_{nn}$ is the optimal solution of the CNN.

$$p(\mathbf{y}_t \mid \mathbf{w}, X) = p(\mathbf{y}_t \mid f_{nn}, \mathbf{w}, X) p(f_{nn} \mid \mathbf{y}_{dclm}^*, \mathbf{w}, X) \tag{10}$$

If $\mathbf{w}$ and $X$ are given and the loss function $\phi_r(\mathbf{y}_t, f_{nn}) = -\frac{1}{2} \sum_l \mid \mathbf{y}_t - f_{nn} \mid^2$, the conditional probability distribution function

$$p(\mathbf{y}_t \mid f_{nn}, \mathbf{w}, X) = \frac{\exp(\phi_r(\mathbf{y}_t, f_{nn}))}{\Xi_1} \tag{11}$$

Meanwhile,

$$p(f_{nn} \mid \mathbf{y}_{dclm}^*, \mathbf{w}, X) = \frac{\exp(-\phi_d(\mathbf{y}_{dclm}^*, f_{nn}))}{\Xi_2} \tag{12}$$

where $\Xi_1$ and $\Xi_2$ are partition functions. Then by maximizing a likelihood function of $p(\mathbf{w} \mid X, \mathbf{y}_t)$ the optimal $w$ can be obtained. In particular, assuming that $\mathbf{w}$ follows Gaussian distribution, we get:

$$C_{\mathbf{w}}(X, \mathbf{y}_t) = arg \max_{\mathbf{w}} [-\frac{\alpha}{2} \parallel \mathbf{w} \parallel^2 + \phi_r(\mathbf{y}_t, f_{nn}) - \log(\Xi_1) - \phi_d(\mathbf{y}_{dclm}^*, f_{nn}) - \log(\Xi_2)] \tag{13}$$

where $\alpha$ is a meta-parameter determined by the variance of the selected Gaussian distributions. Turn it into a minimization problem:

$$C_{\mathbf{w}}(X, y_t) = arg \min_{\mathbf{w}} [\frac{\alpha}{2} \parallel \mathbf{w} \parallel^2 - \phi_r(\mathbf{y}_t, f_{nn}) + \log(\Xi_1) + \phi_d(\mathbf{y}_{dclm}^*, f_{nn}) + \log(\Xi_2)] \tag{14}$$

The iterative optimization algorithm is shown as follows:

## 6 EXPERIMENTAL VERIFICATION

We designed two experiments to verify the effectiveness of the proposed method. The first experiment verified whether the self-explanatory method could improve the interpretability performance

of the CNN without sacrificing its generalization performance. The second experiment verified whether the proposed method can tend towards stability and convergence in the game process.

In the experiments, CNN3(includes 3 convolution layers,3 MaxPooling layers, 3 fully connect layers(FCLs) and 1 output layer),CNN5(includes 5 convolution layers,5 MaxPooling layers, 3 FCLs and 1 output layer), and CNN8(includes 8 convolution layers,8 MaxPooling layers, 3 FCLs and 1 output layer) were used. Traditional training methods on the three types CNN were named as CNN3-Trad, CNN5-Trad and CNN8-Trad respectively. By contrast, our proposed methods on these CNNs were named as CNN3-DCLM, CNN5-DCLM and CNN8-DCLM respectively. All experiments used Mnist(Lecun et al., 1998), FashionMnist(Zalando, 2017), and Cifar-10(Krizhevsky, 2009) benchmark data sets. All algorithms were implemented in Python using the Pytorch library(Paszke et al., 2019). All experiments ran on a server with Intel Xeon 4110(2.1GHz) Silver Processor, 20GB RAM and Nvidia Telsa T4.

**Experiment 1: Performance verification of the proposed method on CNN**. We replaced the discrimination part of three traditionally trained CNNs with soft decision tree(SDT)(Frosst & Hinton, 2017) and designated these methods as CNN3-Trad-SDT, CNN5-Trad-SDT and CNN8-Trad-SDT respectively. The accuracy of CNN, accuracy of STD or DCLM, and interpretation distance corresponding to all methods were shown in Table 1. Some values are "—", which indicates that these results do not exist.

It is observed in Table 1 that the accuracies of all CNN trained by the proposed method are higher than those of the two interpretable models, such as SDT and DCLM, on all benchmark data sets and are around 1.4 percentage points lower than those of all CNN trained by the traditional training method. But it is worth noticing that on the most of data sets the interpretation distances of these CNNs trained by the proposed method are lower than interpretation distances of CNNs obtained by the traditional method. These might prove that that the self-explanatory method can improve the interpretability performance of the discrimination part of a CNN without largely reducing its generalization performance. The accuracies of the DCLMs are higher than those of the SDTs except CNN3-DCLM on FashionMnist data set and CNN3-DCLM on Mnist. These results might prove that the proposed method can find more excellent interpretable model than the traditional method.

**Experiment 2: Convergence test of the proposed method** We designed the experiments to demonstrate convergence of the proposed method. CNN3-Trad, CNN5-Trad and CNN8-Trad were used for comparing with CNN3-DCLM, CNN5-DCLM, and CNN8-DCLM respectively. Every training works out 25 epochs. Experiment results were measured at every epoch and shown in four figures, Fig.2, Fig.3, Fig.5 and Fig.4. Every figure includes nine subplots. The three subplots on the left column were shown for the experiment results on Cifar-10 data set. These subplots on the middle column were for FashionMnist data set and these subplots on the right column were for Mnist data set.

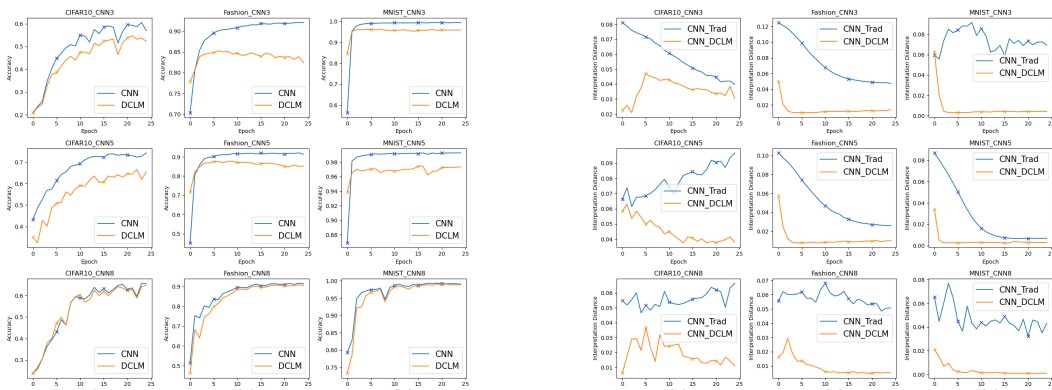

Figure 2: Accuracies of $DCLMs$ and CNNs obtained by the proposed method

Figure 3: Interpretation distance of CNNs from the tradition training method and the proposed method respectively

Table 1: Classification accuracies and interpretation distances.

DATA SET: MNIST

| METHOD | ACCURACY | ACCURACY OF SDT OR DCLM | INTERPRETATION DISTANCE |
|---|---|---|---|
| CNN3-TRAD | 0.971±0.06 | — | 0.005±0.0039 |
| CNN3-TRAD-SDT | — | 0.970±0.017 | — |
| CNN3-DCLM | 0.993±0.006 | 0.958±0.017 | 0.004±0.0010 |
| CNN5-TRAD | 0.993±0.006 | — | 0.005±0.0046 |
| CNN5-TRAD-SDT | — | 0.968±0.017 | — |
| CNN5-DCLM | 0.993±0.007 | 0.973±0.014 | 0.003±0.0010 |
| CNN8-TRAD | 0.984±0.007 | — | 0.003±0.0019 |
| CNN8-TRAD-SDT | — | 0.978±0.012 | — |
| CNN8-DCLM | 0.992±0.009 | 0.989±0.018 | 0.001±0.0010 |

DATA SET: FASHIONMNIST

| METHOD | ACCURACY | ACCURACY OF SDT OR DCLM | INTERPRETATION DISTANCE |
|---|---|---|---|
| CNN3-TRAD | 0.921±0.021 | — | 0.083±0.0355 |
| CNN3-TRAD-SDT | — | 0.842±0.041 | — |
| CNN3-DCLM | 0.920±0.024 | 0.824±0.032 | 0.014±0.0030 |
| CNN5-TRAD | 0.923±0.024 | — | 0.064±0.0119 |
| CNN5-TRAD-SDT | — | 0.792±0.041 | — |
| CNN5-DCLM | 0.921±0.022 | 0.873±0.036 | 0.009±0.0020 |
| CNN8-TRAD | 0.931±0.022 | — | 0.122±0.0397 |
| CNN8-TRAD-SDT | — | 0.785±0.043 | — |
| CNN8-DCLM | 0.914±0.020 | 0.905±0.032 | 0.005±0.0020 |

DATA SET: CIFAR-10

| METHOD | ACCURACY | ACCURACY OF SDT OR DCLM | INTERPRETATION DISTANCE |
|---|---|---|---|
| CNN3-TRAD | 0.681±0.040 | — | 0.044±0.0060 |
| CNN3-TRAD-SDT | — | 0.538±0.057 | — |
| CNN3-DCLM | 0.643±0.038 | 0.601±0.040 | 0.022±0.0030 |
| CNN5-TRAD | 0.744±0.041 | — | 0.036±0.0040 |
| CNN5-TRAD-SDT | — | 0.609±0.038 | — |
| CNN5-DCLM | 0.729±0.037 | 0.684±0.039 | 0.027±0.0040 |
| CNN8-TRAD | 0.754±0.035 | — | 0.040±0.0040 |
| CNN8-TRAD-SDT | — | 0.647±0.035 | — |
| CNN8-DCLM | 0.682±0.038 | 0.661±0.036 | 0.014±0.0030 |

In Fig.2, the accuracies of the DCLMs and the CNNs of every epoch in the game process were shown. From these figures, it is obvious that accuracies of the DCLMs and these CNNs steadily increase in the early stage. In the next stage, these accuracies tend to be stable. This reflects that the game method do not affect the improvement of the generalization performances of these DCLMs and these CNNs. We also find that the accuracy gap obtain by CNN3-DCLM for the FashionMnist data set is much more than other two data sets. Meanwhile, it can be found that the DCLM convergence to a stable state. The main reason is that for CNN3, in the later stage of the game process, new inherent causality implied in the part can not readily be found by the FashionMnist data set. Even so, the proposed method also improves the accuracies of the DCLMs and these CNN-DCLMs steadily. We also find that the gaps between these accuracies of the DCLMs and the CNNs obtained by the CNN8 become very small at the later epochs. This reflects that CNN8 can extract more effective features by which the DCLMs can find more accurate causality implied in the discrimination part of the CNNs.

Fig.3 shows the interpretation distances of the CNNs trained by the traditional method and the CNNs trained by the proposed method. As seen from these subplots, the interpretation distances of CNNs trained by the traditional method are greater than those of the other CNNs at the most of the epochs, especially by the end of the game. The results indicate that the game method can effectively improve the interpretability performance of CNNs. From these subplots in Fig.2and Fig.3, we also

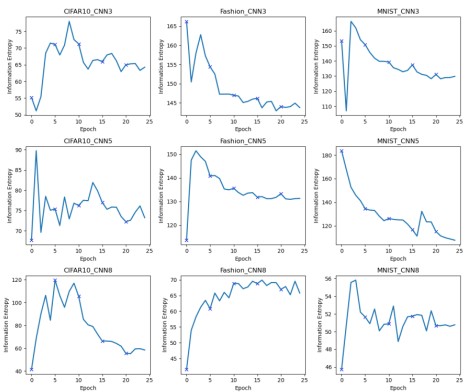

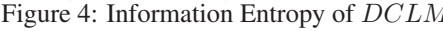

Figure 4: Information Entropy of $DCLM$

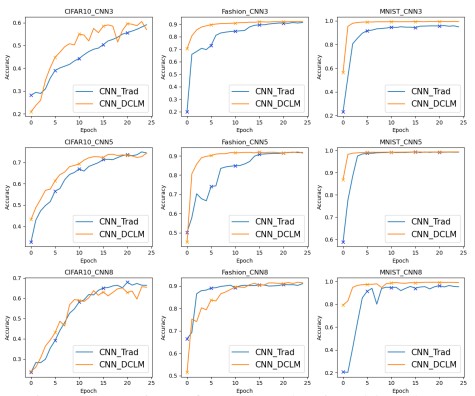

Figure 5: Accuracies of CNNs obtained by the tradition training method and the proposed method

can see that after the 15th epoch, interpretation distances of CNNs from the proposed method tend to converge. The phenomenon indicates that the discrimination part of the CNNs can be explained by its DCLMs at every epoch after the fifteenth epoch.

From Fig.4, it is evident that all DCLMs from CNN3-DCLMs, CNN5-DCLMs, and CNN8-DCLMs have been found to have stable information entropies at the end of the game,which calculate the diversity of disjunction normal forms in the DCLMs obtained by the game method. On Mnist data set, the entropies finally converge 135,113,and 51.3 respectively. On FashionMnist data set, the entropies finally converge 144,132,and 65. On Cifar-10 data set, the entropies finally converge 65,73,and 57. More complicated the extract part of CNN, more small the information entropies of DCLM obtained by the proposed method. The results indicate that the game algorithm can ensure that the diversity of disjunction normal forms of the DCLMs converges to a stable state. The game with the CNN with complex structure can obtain the more robust DCLMs than with the CNN with simply sturcture. The main reason is that the features captured by the CNN with the complex structure is so sparse and robust that the disjunction normal forms of the DCLMs is sparse and robust.

In Fig.5,the accuracies of the CNNs trained by the traditional method and the CNNs trained by the proposed method at every epoch were shown. From these subplots, it can be seen that its accuracies steadily increase in the early stage. But in the following stage, their accuracies tend to be stable and consistent. The main reason is that in the early stage, a tradeoff problem between the generalization performance and interpretability performance of the discrimination part of a CNN inevitably reduces its generalization performance in order to increase its interpretability performance. Though the proposed game method can effectively reduce the gap between two performances, it do not increase the gap between the accuracies of the CNNs trained by the traditional method and the CNNs trained by the proposed method. This reflects that the proposed method is effective for the tradeoff problem.

## 7 CONCLUSION

The performance of the proposed method was demonstrated by experiments on benchmark data sets. The proposed method showed prominent advantages over traditional learning algorithm on CNN for improving the generalization performance and the interpretability performance of the discrimination part of the CNN.

In practical engineering, the proposed method may provide a new learning paradigm. The method can not only predict an relatively accurate result for new input data but also provide a reasonable causal interpretation for the prediction of the discrimination part of a CNN. We suppose that it can solve the black box problem in the discrimination part. We believe that the proposed method provide a way to understand the discrimination part.

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

## A  APPENDIX: SUFFICIENT AND NECESSARY CONDITION FOR CONSISTENT CONVERGENCE OF THE GENERALIZATION PERFORMANCE AND THE INTERPRETABILITY PERFORMANCE

If the minimization of loss function of CNN can guarantee the minimum of its interpretability performance, learning algorithm of the CNN can improve its interpretability performance. If not, the tradeoff problem between the generalization performance and the interpretability performance will exit. For proving the existence of the problem, we focus on a neuron of CNN. From the foot we may judge of Hercules. If an input channel $f(x)$ of the neuron is seen as a kernel function $K(x, w)$($w$ is weight vector including a bias of the neuron), it will span a kernel Hilbert space $\mathcal{H}_K = \{f(x) \in \mathcal{L}_\nu^2(X) \mid f(x) = K(x, w) = \sum_{k=1}^\infty a_k \phi_k(x)\}$ for the neuron. $\mathcal{H}_K$ can be regarded as a linear function set on $\mathcal{L}_\nu^2(X)$. It is a solution space of the neuron. The necessary and sufficient conditions for consistent convergence between the generalization performance and the interpretability performance are discussed below in $\mathcal{L}_\nu^2(X)$ based on the following lemmas.

**Lemma 1.** Continuous linear functional set on a separable Hilbert space $X$ is nowhere dense in a square integrable function space $\mathcal{L}_\nu^2(X)$.

**Lemma 2.** Continuous nonlinear functional set of the separable Hilbert space $X$ is everywhere dense in $\mathcal{L}_\nu^2(X)$.

When the optimal input channel $f^*(x)$ approximate a linear functional in $\mathcal{H}_K$ while the optimal interpretable model $P^*(x)$ don't approximate any linear functional or do not exist in $\mathcal{H}_K$, traditional training process cannot guarantee $f(x)$ approximates $P^*(x)$ according Lemma 1. From Lemma 2, the approximate will cannot converge until $P^*(x)$ approximate $f^*(x)$. Here, if we define the approximation as the similarity between the shapes of function curve of $f^*(x)$ and $P^*(x)$, the sufficient and necessary condition for the consistent convergence of the two performances is $\phi_d(P^*, f^*) = 0$. For the discrimination part of the CNN, the sufficient and necessary condition is still true. However

because of data bias and noisy data in training data set, the condition is difficult to to be ensured in the majority of engineering applications. The tradeoff problem always exists between the two performances of the discrimination part.

According to the above conclusion, in order to completely solve the tradeoff problem, the $\phi_d(P^*, f^*)$(Here $f^*(x)$ is the optimal discrimination part) should be reduced. However $P^*$ and $f^*(x)$ are unknown. Therefore, in training process, extracting a interpretable model $P(x)$ from a discrimination part $f(x)$ and then iteratively reducing $\phi_d(P, f)$ may be a convenient method.

