# OpenReview forum: "A self-explanatory method for the black box problem on discrimination part of CNN"
_ICLR.cc/2021/Conference — Reject_

### Official Review · AnonReviewer3 · 2020-10-26
**The paper is very hard to read.**

**Rating:** 3
**Confidence:** 4

**Review:**

### Paper Summary
The authors proposed Deep Cognitive Learning Model (DCLM) as a surrogate of "discrimination part of CNN" which is the fully connected layers after the convolution layers. The authors claimed that DCLM can explain the causal relationship between the features and the output result of the discrimination part of CNN.

### Quality & Clarity
I found the paper very hard to read. I tried my best to understand the paper. The major contributions of this paper are as follows.
1. The authors proposed to use DCLM to approximates the fully connected layers (FC), i.e. $\mathrm{FC}(x) \approx \mathrm{DCLM}(x)$ for any $x$. Because DCLM is in a logical formula, we can interpret the "discrimination part of CNN" approximately by interpreting DCLM.
2. The authors further proposed to use DCML to "regularize" FC. That is, train CNN so that its FC to be close to DCLM.
The first contribution is described in Sec3, while the second contribution is described in Sec4 and Sec5.

Below, I list the points that makes the paper hard to read.
* The definition of the term "discrimination part of CNN" is described at the beginning of Sec3, although the term is frequently used in Sec1. I could not understand what the "discrimination part of CNN" means when I first read the paper. This problem is fatal because the readers cannot understand the problem the authors want to solve in this study.
* The paper does not describe why interpreting "discrimination part of CNN" is important. Thus, the paper failed to motivate the problem. Although the authors cited some related studies, such as [Wan+20], the importance of the problem need to be described in the paper. This problem is fatal because the readers cannot understand why the problem is important.
* Algorithm1 contains undefined notations. For example, the inputs and the outputs of $\mathrm{CN}$ and $LN$ are not defined. There are no descriptions what $FMs$ and $f_{nn}$ are. Moreover, the second $\mathrm{CN}$ in Algorithm1 outputs $FCMs$ in addition to $FMs$ and $f_{nn}$. This problem is fatal because the readers cannot understand how the proposed method operates.
* The authors merely compared the accuracies of DCLM and its approximation performance. There is no demonstration nor discussion that DCLM can resolve the problem of interpretability of the "discrimination part of CNN", which should be the primal goal of this study. This problem is fatal because the readers cannot understand whether the problem considered in this study is solved.

Overall, I think the paper needs major rewriting so that the main message of the paper to be clear: what the problem is; why it is important; how the authors solved the problem; and how the authors confirmed the problem is solved.

In addition to the readability, I also found some technical errors.
* In Eq.(5), the partition function $\log \Xi$ should appear because it is a function of $y_{dclm}$ and $a$. The authors somehow ignored it.
* In Eq.(13) and Eq.(14), the partition functions $\log \Xi_1$ and $\log \Xi_2$ should appear because they are functions of $w$. The authors somehow ignored them.
There should be some justifications why one can ignore the partition functions. Or, the experimental results need to be updated based on the objective functions with the partition functions.

### Originality & Significance
The use of decision tree for approximating the "discrimination part of CNN" is considered by [Wan+20]. This paper proposes using a logical formula instead of the decision tree. The idea seems to be straightforward, and the innovation made in this paper is marginal.

### Pros & Cons
[Pros]
* I could not find anything positive about this paper.

[Cons]
* The paper is very hard to read. The major rewriting is needed.
* There are some technical errors.
* The improvement over [Wan+20] seems to be marginal.

---

> ### Author Response · Authors · 2020-11-22
> **A self-explanatory method for the black problem on discrimination part of CNN**
>
> Thank you very much for your comments and constructive suggestions. Some responses to the questions have been shown in the following.
>
> (1)	The definition of the term "discrimination part of CNN" is described at the beginning of Sec3, although the term is frequently used in Sec1. I could not understand what the "discrimination part of CNN" means when I first read the paper:
>
> Response:Thank you for your suggestions. We will correct this flaw in subsequent new edition.
>
> (2)	The paper does not describe why interpreting "discrimination part of CNN" is important. Thus, the paper failed to motivate the problem. Although the authors cited some related studies, such as [Wan+20], the importance of the problem need to be described in the paper:
>
> Response:The former part of CNN can be understood as the process of feature extraction. We can obtain the physical interpretation of every neuron through visualization method. However, the fully connected layers cannot be interpreted by visualization methods. Due to the uninterpretable nature of the fully connected layers, it is impossible to know its inherent causality which can conclude the reason for its prediction results. Moreover, it is also impossible to monitor the evolution of the diversity of logical relations implied in the network in the training process in real time, so as to provide the basis for further improving the training process in the future. In the new version, the introduction part will be revised for the flaw.
>
> (3)	Algorithm1 contains undefined notations.
>
> Response:The pseudocode of the algorithm 1 in the first version of the paper was too simple. In the new version the complete pseudocode will be provided.
>
> (4)	The authors merely compared the accuracies of DCLM and its approximation performance. There is no demonstration nor discussion that DCLM can resolve the problem of interpretability of the "discrimination part of CNN", which should be the primal goal of this study.
>
> Response:First of all, the DCLM is an interpretable model, which includes many disjunctive normal forms. These normal forms can be easily expressed as understandable implications. When it keeps approximating and finally reach consistent convergence with the discriminating parts of CNN, the interpretable model can explain the discriminating part of CNN. This point has been proved from both experimental and theoretical perspectives. Finally, we will carefully revise this paper to clarify this point.
>
> (5)	Overall, I think the paper needs major rewriting so that the main message of the paper to be clear: what the problem is; why it is important; how the authors solved the problem; and how the authors confirmed the problem is solved.
>
> Response:Thank you for your suggestions. We will carefully revise this paper to ensure that the latest edition will improve its readability and express the four points more clearly.
>
> (6)	In addition to the readability, I also found some technical errors
>
> Response:Thank you for your reminders. We will carefully revise these equations.
>
> (7)	The use of decision tree for approximating the "discrimination part of CNN" is considered by [Wan+20]. This paper proposes using a logical formula instead of the decision tree. The idea seems to be straightforward, and the innovation made in this paper is marginal.
>
> Response:The innovation of this paper is not only the establishment of interpretable model DCLM, but also the proposal of self-interpretation method, which solves the problem of how to improve the interpretable performance of discrimination part of CNN when there is no prior knowledge and the optimal interpretable model. At the same time, because of the lightweight structure of the interpretable model, it can ensure that the logical relationship in the fully connected layers can be known in time in each epoch of the training process. It is easy to find the change law of this logical relationship to better guide the training process. This can be verified by the experimental results in Fig. 3,4,5,6.
>
> (8)	I could not find anything positive about this paper.
>
> Response: We think that the following points are worth reading in this paper:
> 1. Aiming at the uninterpretable problem of the discrimination part of CNN, this paper proposes a self-interpretation method, which does not need any prior knowledge;
> 2. In order to realize this self-interpretation method, a lightweight interpretable model and interpretation distance are proposed to measure the interpretability performance
> 3. This paper analyzes the black box problem, deduces the necessary and sufficient conditions for this problem, and points out the reasons why the problem cannot be solved at present.

---

### Official Review · AnonReviewer2 · 2020-10-27
**Significant issues in the language severally limit this work, though there are content issues as well.**

**Rating:** 3
**Confidence:** 2

**Review:**

I found this paper very difficult to follow as it has many grammatical and syntactic errors. I believe it needs a significant amount of editing in order for the paper to be published in english. In particular careful attention should be made to omitted particles and pluralization.  This alone is a barrier to publication.

Setting aside the grammatical errors, I believe there are some issues in the content of the paper that would need to be addressed as well in order to make this paper ready for publication.

If I understand the paper correctly, which I am not sure I do, I believe the authors propose a method for extracting an interpretable model which replaces the fully connected layers of a Convolutional Image Network. They present results on toy datasets, MNIST, Emnist, and FashionMNIST.

My primary issue with the paper is that it attempts to provide an 'explainable alternative' to a CNN but this explainable model still relies on the features extracted from the convolutional section of a CNN. The paper does not put forward a convincing argument to justify the focus on the fully connected layers. It is interesting to extract an interpretable model from a fully connected network, but if this is the goal of the paper, then the authors should focus on datasets in which a fully connected network outperforms standard explainable models such as logistic regression but interpretability would still be desirable, such as the MIMIC medical dataset.

The paper would also be significantly improved if more realistic datasets would be explored. The only datasets used are variants of MNIST, in which good performance can be achieved with traditional explainable models.

---

> ### Author Response · Authors · 2020-11-22
> **A self-explanatory method for the black problem on discrimination part of CNN**
>
> Thank you very much for your comments. We will revise the manuscript based on your suggestions.  Some responses to the questions have been shown in the following.
>
> (1)	I found this paper very difficult to follow as it has many grammatical and syntactic errors.
>
> Response:Thank you for your suggestions. We will carefully revise these errors in the latest edition.
>
> (2)	If I understand the paper correctly, which I am not sure I do, I believe the authors propose a method for extracting an interpretable model which replaces the fully connected layers of a Convolutional Image Network.
>
> Response:The purpose of extracting logical network is not to replace the full connection layer of CNN, but to improve the interpretable performance of full connection layer. In the absence of ideal interpretable model, a new self-interpretation method is proposed in this paper. Firstly, the interpretable model is extracted from the full connection layer of CNN, and then the interpretative distance between the two is shrunk by game to improve the interpretable performance of the full connection layer, and to keep its well generalization performance.
>
> (3)	My primary issue with the paper is that it attempts to provide an 'explainable alternative' to a CNN but this explainable model still relies on the features extracted from the convolutional section of a CNN. The paper does not put forward a convincing argument to justify the focus on the fully connected layers. It is interesting to extract an interpretable model from a fully connected network, but if this is the goal of the paper, then the authors should focus on datasets in which a fully connected network outperforms standard explainable models such as logistic regression but interpretability would still be desirable, such as the MIMIC medical dataset.
>
> Response:In this paper, though we extract the interpretable model according to all feature maps, it does not depend on these feature maps but the physical structure of the fully connected layers. The definition of DCLM in paper have provided evidence to show its disjunctive normal forms are related to fully connected layers but not feature maps.
>
> (4)	The paper would also be significantly improved if more realistic datasets would be explored. The only datasets used are variants of MNIST, in which good performance can be achieved with traditional explainable models.
>
> Response:Many thanks. We will revise the manuscript based on your suggestion. Some new data sets will be added for evaluating the proposed method. Moreover, we think that the most important contribution in our paper does not lie in designing the interpretability model but providing a self-interpretation method on CNN’s fully connected layer to improve its interpretable performance. Besides, the experiment in this paper also shows that our method is very excellent.

---

### Official Review · AnonReviewer1 · 2020-10-27
**This paper proposes an interesting framework for training interpretable CNNs, similar to distillation methods. While I found the idea and the framework very compelling, the structure of the paper makes it challenging to evaluate the details of the procedure. Please see my concerns below.**

**Rating:** 5
**Confidence:** 3

**Review:**

This paper proposes an interesting framework for training interpretable CNNs, similar to distillation methods. The authors propose a probabilistic model to approximate CNN predictions (specifically the discriminatory part i.e. fully connected network, and a procedure for training CNN+ DCLM as a game. Results show interesting performance over benchmark datasets in comparison to existing distillation baselines.

Major concerns:
1. While the motivation is easy to understand, notation is continuously introduced in the paper without clear description and definitions. This makes it very challenging to parse the technical correctness of the paper.

2. What is the iterative optimization procedure described in Page 5? Highly unclear how the CNN model is actually updated in Algorithm 1 to do the minimization of Eq (14)

3. Can the authors elaborate why the accuracy gap for more challenging datasets like FashionMNIST is much more than simpler datasets for the DCLM model? I also suggest authors to include other datasets instead of focusing on MNIST only.

4. The entropy analysis is interesting but I would've liked to see more insight into why entropy behavior across different datasets is so variable.

5. Way too many typos in the paper. Please proof-read and correct, I only highlighted a few:
 1. Update_CNN_Gradient in Algorithm 1
 2. Information Entropy in Figure 5 caption
 3. Clarify what "discrimination part" of CNN means, very early on in the introduction and abstract.

6. Did the authors do a qualitative analysis of where exactly the approximate DCLM has lower accuracy compared to the CNN model and why?

---

> ### Author Response · Authors · 2020-11-22
> **A self-explanatory method for the black problem on discrimination part of CNN**
>
> Thanks for the thoughtful reviews and constructive suggestions. We will revise the manuscript based on your suggestions.  Some responses to the questions have been shown in the following.
>
> (1)	While the motivation is easy to understand, notation is continuously introduced in the paper without clear description and definitions.
>
> Response: In the paper, some notations are not described and defined in the first appearance. In its’ new version, we will make revisions for the negligence.
>
> (2)	What is the iterative optimization procedure described in Page 5? Highly unclear how the CNN model is actually updated in Algorithm 1 to do the minimization of Eq (14).
>
> Response: Thanks for your suggestions. The pseudocode of the algorithm 1 in the first version of the paper was too simple. In the new version the complete pseudocode is provided.
>
> (3)	Can the authors elaborate why the accuracy gap for more challenging datasets like FashionMNIST is much more than simpler datasets for the DCLM model? I also suggest authors to include other datasets instead of focusing on MNIST only.
>
> Response: We think that it is chiefly because some challenging datasets can make input neurons capture more features and cause more intricate logic relationships between the activated state and the inactivated state of input neurons. In this situation, the logical relationship obtained by our game maybe not a fully complete one. But the main contribution of this paper is to provide a self-explanatory learning paradigm, which can real-time obtain the inherent causality of the full connection part of CNN without any prior knowledge. More significantly, our work improves its interpretable performance based on ensuring its generalization performance does not obviously decline. Further studies will further optimize the method.
>
> (4)	The entropy analysis is interesting but I would've liked to see more insight into why entropy behavior across different datasets is so variable.
>
> Response: Information entropy calculates the diversity of logical relations obtained by game from full connection part. However, in the initial epoch, the diversities of logical relations obtained by different training sample sets are different, so its initial information entropy values are also different, leading to different information-entropy curves. However, no matter what kind of structure of fully connected parts, this method can always ensure that the entropy tends to a smaller region, which shows that the method can ensure the diversity of logical relations tend to be stable. At the same time, through the change of entropy, we can perceive the change law of the inherent logical relationship of the fully connected parts. We will carefully revise this paper to ensure that the latest edition will improve its readability and express these contents more clearly.
>
> (5)	Way too many typos in the paper. Please proof-read and correct, I only highlighted a few.
>
> Response: Thanks for your comments, we will scrutinize this paper and correct all mistakes.
>
> (6)	Clarify what "discrimination part" of CNN means, very early on in the introduction and abstract.
>
> Response: we will add the meaning of "discrimination part" in the introduction and abstract.
>
> (7)	Did the authors do a qualitative analysis of where exactly the approximate DCLM has lower accuracy compared to the CNN model and why?
>
> Response: We think that this is mainly attributed to that data bias and noisy data in training data set make it difficult for DCLM to approximate to the optimal fully connected part of CNN, so the generalization performance of DCLM is worse than CNN. In its’ new version, we will express these contents more clearly.

---

### Decision · Program_Chairs · 2021-01-07
**Final Decision**

**Decision:**

Reject

**Comment:**

In this paper, the authors work on improving the interpretability of CNNs following distillation methods. The paper is written in such a convoluted way and with many changes in the notation that makes it hard to understand what they are proposing and what for. This is a major impediment for the paper going forward. But also, the reviewers point out some clear aspects of the paper and the interpretability that it provides for CNNs, for example, the entropy analysis and its variability or referring to the discrimination part of the CNN or mentioning an explainable alternative to CNNs, while still relying on the convolutional filters. The authors need to dedicate more time to explain this paper carefully before it can be properly reviewed and published at any major conference.